# Resveratrol Analogues as Dual Inhibitors of Monoamine Oxidase B and Carbonic Anhydrase VII: A New Multi-Target Combination for Neurodegenerative Diseases?

**DOI:** 10.3390/molecules27227816

**Published:** 2022-11-13

**Authors:** Simone Carradori, Marialuigia Fantacuzzi, Alessandra Ammazzalorso, Andrea Angeli, Barbara De Filippis, Salvatore Galati, Anél Petzer, Jacobus P. Petzer, Giulio Poli, Tiziano Tuccinardi, Mariangela Agamennone, Claudiu T. Supuran

**Affiliations:** 1Department of Pharmacy, “G. d’Annunzio” University of Chieti-Pescara, Via dei Vestini 31, 66100 Chieti, Italy; 2Neurofarba Department, University of Florence, 50019 Sesto Fiorentino, Italy; 3Department of Pharmacy, University of Pisa, Via Bonanno 6, 56126 Pisa, Italy; 4Pharmaceutical Chemistry and Centre of Excellence for Pharmaceutical Sciences, North-West University, Potchefstroom 2520, South Africa

**Keywords:** resveratrol analogs, phenols, monoamine oxidases, carbonic anhydrases, molecular modeling, neurodegenerative diseases

## Abstract

Neurodegenerative diseases (NDs) are described as multifactorial and progressive syndromes with compromised cognitive and behavioral functions. The multi-target-directed ligand (MTDL) strategy is a promising paradigm in drug discovery, potentially leading to new opportunities to manage such complex diseases. Here, we studied the dual ability of a set of resveratrol (RSV) analogs to inhibit two important targets involved in neurodegeneration. The stilbenols **1**–**9** were tested as inhibitors of the human monoamine oxidases (MAOs) and carbonic anhydrases (CAs). The studied compounds displayed moderate to excellent in vitro enzyme inhibitory activity against both enzymes at micromolar/nanomolar concentrations. Among them, the best compound **4** displayed potent and selective inhibition against the MAO-B isoform (IC_50_ MAO-A 0.43 µM vs. IC_50_ MAO-B 0.01 µM) with respect to the parent compound resveratrol (IC_50_ MAO-A 13.5 µM vs. IC_50_ MAO-B > 100 µM). It also demonstrated a selective inhibition activity against hCA VII (*K*_I_ 0.7 µM vs. *K*_I_ 4.3 µM for RSV). To evaluate the plausible binding mode of **1–9** within the two enzymes, molecular docking and dynamics studies were performed, revealing specific and significant interactions in the active sites of both targets. The new compounds are of pharmacological interest in view of their considerably reduced toxicity previously observed, their physicochemical and pharmacokinetic profiles, and their dual inhibitory ability. Compound **4** is noteworthy as a promising lead in the development of MAO and CA inhibitors with therapeutic potential in neuroprotection.

## 1. Introduction

Neurodegenerative diseases (NDs) are chronic and multifactorial diseases and involve physiological, biochemical, and chemical changes, mediated by different activation pathways [1]. This leads to the progressive loss of neurons and neuronal connections in the central nervous system (CNS), which normally leads to cognition and motor dysfunction [2,3]. The World Health Organization (WHO) and United Nations (UN) statistics report that the population is aging, and this results in a significant loss of work productivity and enormous economic costs to society in managing aging illnesses such as NDs [4]. Research projects in this area are very different in approach and disease focus, ranging from genetic and molecular biological studies to computational modeling. Recent discoveries have identified a spectrum of distinct pathways that have been altered, and these pathways serve as a basis for a better understanding and targeted research [5].

Oxidative stress seems to be a major factor in ND pathogenesis and progression [6]. In this context, monoamine oxidases (MAOs) have been considered to be one of the most important factors. In mammals, two distinctive MAO enzymes exist, namely MAO-A and MAO-B, which are distinguished by their substrate and inhibitor selectivities [7]. The MAOs bind tightly to the outer mitochondrial membrane of glial, neuronal, and other cells, and regulate biogenic amines by oxidative deamination: MAO-A preferentially metabolizes epinephrine, norepinephrine, and serotonin and is selectively inhibited by clorgyline; MAO-B is selectively inhibited by selegiline or rasagiline and specifically deaminates β-phenethylamine [8]. As a source of oxidative and inflammatory stress, the MAOs have attracted much attention in recent years. In this respect, enzymatic turnover is associated with the production of hydrogen peroxide and reactive aldehydes [9,10]. Selective MAO-A inhibitors are widely applied in the therapy of depression [11], while selective MAO-B inhibitors have been employed alone or in combination therapy to treat Alzheimer’s and Parkinson’s diseases [12,13,14,15].

Concurrently, several studies shed light on carbonic anhydrases (CAs) as possible new targets for Alzheimer’s disease (AD) treatment [16,17]. CAs are metalloenzymes catalyzing the reversible hydration of carbon dioxide to bicarbonate and a proton; some of these enzymes have been identified to play an antioxidant role in cells during oxidative processes [18]. The CA isoforms are classified based on various properties, such as catalytic activity, tissue distribution, expression levels, subcellular location, kinetic properties, and inhibitor sensitivity. To date, at least 16 CA isoforms have been identified in mammals [19]. Oxidative-induced protein modifications may alter their functions, including their catalytic activity [18,20]. Despite the localization and high expression of CAs in the brain, their functions are still not fully understood. CAs play a pivotal role in physiological processes such as brain pH control, neuronal excitability, and cognition, and they impact animal learning in different spatial models [21] and aversive [22] and object recognition [23] memories [24]. Indeed, infusion of CA modulators in specific areas, such as the hippocampus and the frontal cortex, affected fear memory extinction and social discrimination [25]. Among the human isoforms, CA II has been identified to associate with numerous abundant plaque proteins, suggesting that it may play a central role in plaque development or co-occur with plaque formation [26]. The high CA II levels found in central and in peripheral systems also suggest the possibility that CA II expression may represent a biomarker for cognitive disorders [27]. Furthermore, promising preclinical evidence using CA inhibitors (CAIs) or activators (CAAs) in experimental models has been reported, with a special focus on CA VII [28,29,30]. In addition, the inhibition of mitochondrial CAs (CA VA and VB) could be useful in protecting against oxidative stress, which may lead to a slowing down in the progression of NDs [31,32].

A significant effort is being made to obtain drugs to manage the NDs in an efficacious way [33,34]. Considering the complexity of NDs, drugs with multi-target activities might be useful in the treatment of such diseases. Thus, a more appropriate therapeutic strategy based on the multi-target-directed ligand (MTDL) paradigm could be applied to design compounds against NDs [35,36]. In recent times, attention has shifted to improving lifestyle habits with the help of preventive health care supplements. Several studies in the last few years have focused on the therapeutic potential of natural compounds for NDs, especially those derived from plant extracts [37,38]. Their limited side effects and their multiple target mechanisms of action make the use of natural compounds particularly attractive for the prevention or treatment of multifactorial diseases, such as those that produce neurodegeneration [39].

Resveratrol (RSV, 3,4′,5-trihydroxystilbene, Figure 1) is a stilbene, a member of a subclass of phenolic compounds found in several plants, including grapes, blueberries, raspberries, and peanuts, and in red wine [40,41]. Its multiple biological activities range from cardiovascular diseases [42] and antiaging [43,44] to antimicrobial [45,46,47] and radioprotective effects [48] and benefits on bone health [49]. In particular, the neuroprotective effects of RSV have been investigated in several in vitro and in vivo experimental models [50,51,52]. Due to its antioxidant properties [53], RSV has received increasing attention in preventing various pathologies associated with oxidative stress, such as aging and neurodegenerative pathologies. Recent lines of evidence suggest that RSV can directly target multiple signaling cascades involved in NDs, such as anti-inflammatory activity and the inhibition of the aggregation of toxic Aβ amyloid protein [54,55]. Due to its ability to simultaneously interact with several targets, RSV could be used as an MTDL [56,57,58], which may be attributed to the stilbene core, a privileged scaffold that represents a good starting point in the design of multifunctional drugs for NDs [59,60].

Some studies have also shown that RSV derivatives act as MAO-B inhibitors [61,62]. MAO inhibitors have received increasing interest for their roles in monoamine neurotransmitter metabolism and oxidative stress, as well as neuroprotective effects against NDs [63,64]. On the contrary, not many studies have been published concerning the inhibitory activity of RSV and its derivatives against CA. Unfortunately, in spite of the benefits of RSV, this compound shows, in mammals, very low bioavailability because of rapid liver metabolism and low water solubility [65,66]. To solve this problem, several studies have been performed to obtain synthetic analogs with improved pharmacokinetic properties while retaking desirable pharmacodynamic characteristics. This prompted numerous research groups to investigate innovative synthetic derivatives and/or the development of nanoformulations [67,68,69,70] with optimized properties [71,72,73]. Numerous studies have also been conducted on hybrids and derivatives of RSV, and their activities have been reviewed [55,74,75,76].

In continuation of our research for unraveling the potential of novel stilbene-based compounds, we focused on the substitution of the 3,5-dihydroxybenzene ring and maintaining the 4′-OH group due to its well-documented role in the antioxidant activity [77]. Previous results showed that the introduction of a substituent with different electronic and lipophilic properties on the 4-position of aromatic ring A (R_1_) improved both the anticancer [78,79] and antibacterial [80] activities. In a recent study, we reported a set of diverse molecules (compounds **1**–**9**, Figure 1) in which the resorcinol group of RSV was replaced by a substituent on the 2- and/or 4-position (ring A, compounds **1**–**7**) or substituted by a naphthyl or a pyridyl moiety (compounds **8** and **9**). All these compounds were evaluated for their ability to modulate the vitality of the C2Cl2 cell line, and the most active compounds were also tested for antioxidant activity. It should be noted that the presence of the 4-trifluoromethyl group and the 3′-chlorine promoted the proliferative capacity of the cells, probably due to the remarkable antioxidant activity and significant reduction in superoxide anion levels, which are more pronounced than the corresponding activities recorded for RSV [81]. Moreover, the presence of halogens induced an increase in lipophilia by improving the *logP* value of all compounds to around or above 5.0, making them likely orally active drugs in humans.

Building on these preliminary results, in this work, we aimed to explore the multi-target activities for the same RSV derivatives in an attempt to optimize their pharmacological profiles. Here we report the dual inhibitory effect of RSV derivatives on hCAs and the MAOs. This is the first study that investigates the dual activity of this scaffold on hCAs and MAOs, while other RSV-based hybrids have been reported as dual inhibitors of MAOs and cholinesterases to treat neurodegenerative diseases [56,74].

After in vitro experimentation, the possible mechanism of interaction of **1–9** with the MAOs and CAs was also investigated by structure-based computational studies. The compounds were subjected to molecular docking and dynamics simulations, which highlighted the structural requirements useful for the design of dual inhibitors.

## 2. Results and Discussion

### 2.1. Chemistry

In order to evaluate the multiple effects of the presence of different substituents with lipophilic or electronic properties on different targets involved in NDs, the previously synthesized/home-made compounds **1***–***9** were prepared. They were obtained following reported procedures [78,81]. After purification, they were characterized by ^1^H and ^13^C-NMR spectroscopy; the geometry of the double bond was established by the J-value range from 15 to 16 Hz of the trans-olefinic proton with respect to cis-stilbene olefinic protons from 7.4 to 8.6 Hz for the double of doublet signal (dd) for double bond hydrogens reported in the literature [82].

### 2.2. Biology

#### 2.2.1. MAO Inhibition Study

To investigate the structure-activity relationships (SARs) of these RSV derivatives and find optimal candidates for further development, we expanded the knowledge regarding their ability to modulate enzymatic pathways involved in NDs, such as those pathways catalyzed by the MAOs and CAs.

Starting from the evidence that RSV and some of its derivatives inhibit the MAOs [74,83,84,85,86,87], our interest was to explore the activities of our RSV-based phenols **1**–**9** on the same target. MAO inhibition activity has been confirmed in the tested compounds using the previously reported procedure [88]. The IC_50_ values of the compounds against both A and B isoforms are summarized in Table 1 and compared to harmine and isatin [89]. 

Analysis of the results shows that all RSV derivatives inhibit the MAOs in the low micromolar range and are more potent than RSV, used as a reference. All the compounds exhibited IC_50_ values in the range of 0.43 to 21.3 µM for MAO-A and 0.011 to 14.2 μM for MAO-B, showing a very interesting selectivity toward the B isoform (except for compound **5**) which is comparable to or higher than that of isatin, a natural oxidized indole that is a marker of stress. The compound substituted with the naphthalene ring (**8**) showed MAO-B-selective inhibitory activity (IC_50_ value of 0.38 μM vs. IC_50_ of 2.20 μM for MAO-A, MAO-B selectivity index of 5.8), while the compound substituted with pyridyl (**9**) showed slightly poorer MAO inhibitory activities (IC_50_ values of 21.3 and 11.9 μM for MAO-A and MAO-B, respectively) (Table 1). A similar activity trend was also observed for compounds **1**–**3** for which the number of chlorine atoms affected the activity, while the presence of this halogen on ring B (at position 3′) was preferred. Moreover, lower IC_50_ values were obtained for compound **4** (IC_50_ MAO-A 0.43 μM vs. IC_50_ MAO-B 0.011 μM, MAO-B selectivity index of 38). This compound is thus a more potent inhibitor than RSV (IC_50_ MAO-A 13.5 μM vs. IC_50_ MAO-B > 100 μM). In this case, the introduction of trifluoromethyl in the 4-position and a chlorine atom in the 3′-position led to a significant increase in the inhibition activity, while in other cases the presence of a 3′-chlorine (alone in compound **1** or with the 2,4-dichlorobenzene moiety in compound **2**) does not significantly affect the IC_50_ values. Comparing the potency of compounds **1** vs. **4** and **4** vs. **5**, it is evident that the addition of a Cl substituent on the 3′-position affects MAO-B activity much more than the introduction of a CF_3_ group on the 4-position. The substitution of the OH group of RSV with CN or OCH_3_ led to a slight improvement in inhibitory activity and B-selectivity. The resveratrol analogs of this study do not contain functional groups that could result in the irreversible inactivation of the MAO enzymes. It is thus reasonable to suggest that the study compounds are reversible MAO inhibitors. Furthermore, resveratrol has been reported to inhibit MAO with a competitive mechanism [90].

#### 2.2.2. CA Inhibition Study

CA isoenzymes play important roles in many biochemical and physiological processes [17]. Specific inhibitors of hCA I and II isoenzymes have been used for the treatment of several diseases in the clinic (e.g., as diuretic and anti-glaucoma agents) [19,30]. To the best of our knowledge, only two studies have been conducted with RSV as an inhibitor of CA using the stopped-flow technique or the esterase activity assay [91,92]. In this study, we determined the effect of the new RSV derivatives on the inhibition of seven physio-pathologically relevant CAs. As the reference inhibitor, acetazolamide (AAZ) was used.

The tested phenol compounds exhibited inhibition in the micromolar range. RSV was a discrete pan-isoform inhibitor with a preference for hCA IX and XII. Only compounds **1**, **2**, and **4** inhibited a greater number of isoforms (Table 2). The following SARs were noted: Only compounds with a chlorine atom on the 3′-position exhibited *K*_I_ values in the low micromolar/submicromolar range (<5 µM). Compounds **1** and **2** were the best inhibitors toward the CA XII isoform (*K*_I_ 2.7 µM and 4.5, respectively), while compound **4** resulted in nanomolar inhibition and higher selectivity for the CA VII isoform (*K*_I_ 0.7 µM). Compared to the other compounds tested, compound **4** is noteworthy as the most potent inhibitor with the best selectivity. Even if its CA inhibition ability is not higher than that of reference compounds, its capability to block both the tested enzymes involved in NDs suggests that **4** could serve as a starting point for further studies. Among all tested compounds, it is the only one able to block the two enzymes in a selective manner, suggesting the possibility of a potential dual action. Since the effect of the kind of substituents could not be sufficient to explain its influence on activity, to better understand the reasons for this selectivity, the interactions at the molecular level have been studied.

### 2.3. Computational Studies

#### 2.3.1. Monoamine Oxidase and Compounds **1**–**9**

A structure-based computational study was conducted on all the synthesized compounds to understand how they fit within the MAO-A and MAO-B active sites. The robustness of the docking protocol is demonstrated by the reproduced geometry of both crystallographic ligands (harmine in MAO-A) [93] and safinamide MAO-B [94] (see Appendix A), with RMSD of 0.1364 Å and 0.3850 Å for MAO-A and MAO-B, respectively.

It is well known that the MAO-A and MAO-B binding sites are distinct because of their different dimensions and shapes: while the MAO-A active site is slightly larger and shorter, the MAO-B active site is more elongated [95], as reflected by the shape of their selective inhibitors harmine and safinamide. To elicit the MAO-A and MAO-B binding site differences, SiteMap analysis was carried out. This program maps the protein surface to identify and characterize binding regions, evaluating their dimensions and properties. The calculations performed on the two enzymes shows similar site scores (Table 3), although MAO-B has a larger pocket. In particular, the two active sites are characterized by a high hydrophobicity (3.12 and 3.474, respectively), which shows that they can bind small ligands endowed with a hydrophobic character such as the stilbenol derivatives.

In MAO-B, the compounds’ stilbene moiety occupies the active site’s bipartite hydrophobic region. However, substitution of the hydroxyls in 3- and 5-position of RSV with more hydrophobic functions produces a better fit of the lipophilic groups in the hydrophobic area of MAO-B than MAO-A.

Focusing on the most active compound, we observed that compound **4** establishes hydrophobic interactions between the stilbene scaffold and residues L164, L167, F168, Y326, and L328 and the FAD in the hydrophobic cage, while an H-bond is established between the backbone carbonyl oxygen of P102 and the hydroxyl functional group of the ligand (Figure 2A,B). Interestingly, the docking score of **4** is similar to that obtained for safinamide (−10.336 vs. −10.375), which possesses an IC_50_ of 98 nM [96]. The Cl substituent on the 3′-position of compound **4** is placed in the same region of the enzyme occupied by the fluorine substituent of safinamide, suggesting a positive contact in that region for halogen atoms.

All other ligands bind in the active site of MAO-B in a very conserved way, with the stilbene scaffold aligned and establishing contacts in the hydrophobic portion of the active site. Considering the docking pose with the best docking score, the OH of compounds **2**–**3** and **6**–**7** forms an H-bond contact with the P102 carbonyl oxygen similarly to **4**, while for compounds **1** and **9,** the hydroxyl group is oriented toward the FAD. Noteworthily, when considering the five best docked poses, due to the symmetry of the molecules, all compounds exhibit docking orientations with the OH group facing FAD or interacting with P102.

In MAO-A, the *E*-stilbenol derivative **4** occupies the hydrophobic cage (F352, Y407, Y444) and establishes π–π interactions between the phenyl rings and F208 and Y407 and a halogen–π interaction between the chlorine of **4** and Y197; finally, the hydroxyl group is oriented towards the FAD (Figure 3A,B). The presence of the 3′-chlorine and the 4-trifluoromethyl increases the ligand’s hydrophobic area, which is better accommodated in the elongated pocket of the MAO-B active site than in the shorter site of MAO-A. In addition, the docking score of the best pose of compound **4** suggests a certain degree of selectivity for isoform B (−8.244 in MAO-A; −10.336 in MAO-B).

All other ligands show the same π–π interactions of the stilbene moiety with F208 and Y407 in the active site of MAO-A. The hydroxyl group of compounds **1**, **2**, **7**, and **8** is oriented in the same direction as for **4**, while for compounds **3**, **5**, **6**, and **9**, it projects in the opposite direction, away from the FAD. Moreover, compound **2** makes an additional H-bond contact with the backbone of G443, while compound **8** interacts with T336. Considering the first five docked poses in MAO-A, all compounds (except for **1** and **4**) can dock in a reversed geometry with the OH group placed in front of or opposite to FAD. This is similar to what has been observed for MAO-B.

Focusing our attention on compound **4,** which was not only the most potent toward MAO-B and MAO-A but also the most B-selective, it may be concluded that the MAO-A/MAO-B selectivity can be predicted by the docking analysis, which revealed a better fit in the MAO-B site, as shown by the docking scores for binding to the two enzymes. However, docking calculations do not provide an explanation for the increased MAO-B inhibition activity of this compound compared to its close analogs. To obtain more insight into these data, we carried out a 100 ns MD simulation on the MAO-B complexes with compounds **4** and **5**. Compound **5**, the homolog lacking the 3′-Cl substituent of **4**, presents a three orders of magnitude lower activity with respect to the lead, but scored similarly with the docking calculations (−9.617 and −10.336, respectively) and exhibited a similar docked geometry.

In the MD simulation of ligand **4** bound to the MAO-B binding site, the ligand is subjected to minimal movement with a maximum displacement that does not reach 1 Å. In addition, the protein is relatively stable, reaching a maximum RMSD of 2.25 Å (Figure 4A). The analysis of the protein RMSF graph confirms that the ligand is stabilized by interacting residues. Most ligand–protein contacts are hydrophobic, as predicted by docking studies. The residues involved are L171, Y326, Y398, and Y435. Water-mediated contacts are established between the ligand’s OH group and P102 and I199 (Figure 4C). The ligand is almost completely embedded in the binding site with very limited solvent exposure.

In the MAO-B complex with compound **5**, the movement of both ligand and protein are more marked, with the ligand reaching an RMSD of 4 Å. The RMSF profile of the protein presents with a more pronounced perturbation of the terminal residues, while the rest of the protein maintains a low RMSF value (Figure 5B). In addition, contacts between ligand 5 and the protein are mainly hydrophobic and involve Y326, I199, L171, and Y398, while a stable H-bond is formed with P102, which is in part water-mediated (Figure 5C,D).

The MD analysis suggests that the higher potency shown by compound **4** can be attributed to its better fit in the binding site, and the stabilizing effect on the protein when compared to ligand **5**, as demonstrated by its low RMSD and the RMSF profile shown by the protein.

To obtain more insight into the energies that drive the recognition process, the MD trajectory was exploited to calculate the binding ∆G of the two complexes (Table 4) by MM-GBSA. Even though calculated absolute values cannot be directly extrapolated to experimental data, the obtained energies correlate with the measured IC_50_, with the most active compound presenting a lower ∆G. The analysis of contributing factors to the overall final value highlights the critical contributions of the van der Waals, lipophilic, and π–π stacking contacts, while ligand **4** pays more significant desolvation energy compared to ligand **5**.

These results are in agreement with what has already been observed in the literature, where a correlation between lipophilicity and MAO-B inhibition has been identified, particularly for MAO-B inhibition [97]. Moreover, estimated values present a good accordance with calculated ∆G values for selegiline and rasagiline binding MAO-B [98,99]. As compound **4** is the most potent against MAO-A, the MM-GBSA binding ∆G has also been calculated for this system. Obtained results are in line with experimental IC_50_ values. Moreover, the analysis of energy contributions to the global binding energy highlights the lower effect of lipophilic and van der Waals energy in the binding of compound **4** within MAO-A counterbalanced by the reduced desolvation energy paid by the ligand and protein in the recognition process.

#### 2.3.2. Carbonic Anhydrases and Compounds **1**–**9**

With the objective of predicting binding modes and analyzing the ligand–protein interactions of the RSV derivatives within the hCA isoforms, molecular modeling studies including a robust docking procedure, followed by molecular dynamics (MD) simulations in an explicit water environment, were carried out. This study may provide more insight into the selectivity of the RSV derivatives for hCA VII compared to the other hCAs. Compound **4**, which showed the highest potency for the inhibition of hCA VII, as well as the most potent inhibition of the MAOs, was the focus of this study as a representative ligand of the series. This compound was thus docked into hCA VII using the GOLD software, and the predicted ligand–protein complex was subjected to 50 ns of MD simulation studies (see Section 3 for details). Initially, our modeling studies focused on addressing a specific type of binding mode for compound **4**, in which the ligand formed water-mediated interactions with the catalytic zinc(II) ion, in agreement with the binding mode experimentally observed for small phenolic compounds [100]. However, such studies led to inconclusive results. In fact, the ligand–protein complexes predicted based on this binding hypothesis were found to be unstable and inconsistent with the SAR data. For this reason, we decided to explore a more reliable binding hypothesis that could also provide insights into SAR data.

As shown in Figure 6, compound **4** is predicted to bind to the catalytic site in a different orientation compared to the classic sulfonamide inhibitors [101]. In fact, the ligand is bound to the protein with its CF_3_ group placed in proximity of the zinc-binding pocket, but without directly interacting with the ion, while the chlorophenol moiety is positioned in the solvent-exposed region of the binding site. Specifically, the ligand occupies a rather narrow pocket defined on one side by Q94 and F133, with the latter possibly establishing a T-shaped stacking with the chlorophenol ring of the ligand. The other side of the pocket is defined by N64, Q69, and D71 and is supported by a network of H-bonds formed among these residues, which thus represents a key element for the binding stability of the ligand. Compound **4** is anchored to the pocket by forming H-bonds with N64 and T202 through its trifluoromethyl moiety, while the phenol group of the ligand forms a charge-assisted H-bond with K93. Based on the predicted orientation, the inhibitory activity of the ligand is dependent on steric hindrance generated by its binding to this specific area of the catalytic site, as observed for coumarin and thiocoumarin derivatives, which exert their inhibitory activity without directly interacting with the catalytic zinc ion [102] (Figure 6).

Figure 7 shows the predicted **4**–hCA XII complex. The orientation of the ligand is comparable to that observed in hCA VII, since the trifluoromethylphenyl group maintains the same interactions established with N64 and T202 in hCA VII (N92 and T227 in hCA XII, respectively). Within the water-exposed region of the binding site, residue K93 that is present in hCA VII is replaced by T116, which still allows for the formation of an H-bond with the hydroxyl group of the ligand (although not charge-assisted as in hCA VII); nevertheless, the lateral sides of the pocket exhibit some differences that could affect the stability of the ligand interactions. In particular, the interaction network among N64, Q69, and D71 observed in hCA VII is not conserved due to the replacement of Q69 with K97 and of D71 with N99. The presence of the highly mobile K97 does not allow the formation of stable H-bonds with the surrounding residues, which are necessary to maintain the narrow shape of the binding pocket, a contributing factor to the stability of the ligand. Furthermore, the replacement of A157 for F133 in hCA VII results is the absence of the T-shaped stacking interaction that is established between the chlorophenol moiety of the ligand and F133 in hCA VII. These considerations may explain the reduced inhibitory potency of **4** toward hCA XII compared to hCA VII (Table 2).

An analysis of the predicted ligand–protein interactions in the binding site of hCA II (Figure 8) shows that the two H-bonds between the trifluoromethyl moiety and the side chains of N62 and T199 (corresponding to N64 and T202 of hCA VII, respectively) are maintained. On the other hand, the chlorophenol ring cannot establish any H-bonding due to the presence of I91 that substitutes K93 of hCA VII and T116 of hCA XII. The shape of the binding pocket appears to be similar, although the residues involved in the H-bond network are different from those present in hCA VII. In particular, the network is formed among N62, N67, and E69, the homologous residues to N64, Q69, and D71 of hCA VII, respectively. However, the orientation of the ligand does not allow for the formation of interactions with the conserved residue F130 (homologous to residue F133 in hCA VII), which could further stabilize its binding mode, due to the presence of the bulky side chain of I91 and the lack of H-bond interactions with the solvent-exposed region of the active site. These considerations are in agreement with the inhibition potency of the ligand against hCA II, which was found to be about 22-fold weaker than that against hCA VII.

Figure 9 shows the binding mode of the ligand in hCA IX, which is comparable to that observed in hCA II. The hydrophobic residue L223 replaces K93 of hCA VII, and thus the H-bond interaction with the hydroxyl group of the ligand (as observed in hCA II) is absent. This replacement also results in a slight enlargement of the pocket. Moreover, F133 present in hCA VII is replaced by V262, which cannot form the T-shaped stacking interaction with the ligand. Finally, the replacement of D71 in hCA VII for T205 in hCA IX may decrease the rigidity of the pocket due to its weak interaction with Q203. Nevertheless, the ligand adopts a binding mode similar to that observed in hCA II, which is consistent with its comparable potencies against hCA IX and II, and with its reduced activity against these two isoforms with respect to hCA VII, as reported in Table 2.

The results of the docking studies performed for compound **4** into hCA I, VA, and VB highlighted that the ligand is not able to interact with these three hCA isoforms adopting a binding mode similar to that predicted into hCA VII; this is consistent with the weak inhibitory activities experimentally observed for compound **4** against hCA I, VA, and VB, which were found to be up to 120-fold lower than that observed against hCA VII (Table 2). By inspecting the catalytic site of hCA I, it is evident how the presence of non-conserved residues with respect to hCA VII is not compatible with the ligand orientation observed in this latter isoform. In fact, the presence of H201 and V63 replacing T202 and N64 in hCA VII, respectively, determines the lack of the two H-bond interactions anchoring the trifluoromethyl group of the ligand to the inner portion of the binding site (Appendix A). In addition, there is high variability in the whole pocket occupied by the ligand, where the pattern formed by N64, Q69, and D71 in hCA VII is replaced by V63, H68, and N70, respectively; these residues cannot form the H-bond network observed in hCA VII, thus determining a lower rigidity of the binding pocket. Moreover, the presence of H68 determines a drastic remodeling of the binding pocket shape, filling part of the excluded volume that should be occupied by the ligand. These considerations are consistent with the low inhibitory potency of the ligand against hCA I observed through experimental assays. A similar situation can be observed in hCA VA and VB. In hCA VA, the presence of W36 in place of T62 of hCA VII generates a steric hindrance impeding the proper orientation of the ligand; in fact, W36 pushes the adjacent residues toward the binding pocket, thus filling part of the volume that should be occupied by the ligand (Appendix A). Furthermore, T38 replacing N64 present in hCA VII prevents the ligand from establishing an H-bond interaction with its CF_3_ group. Moreover, residue E45, due to its longer side chain with respect to the homolog D71 in hCA VII, protrudes toward and interacts with K93; both residues thus occupy the free space where the chlorine atom of the phenolic ring of the ligand should be positioned, thereby further decreasing the excluded volume available to the ligand. Finally, the analysis of the catalytic site of hCA VB shows that, as observed for hCA VA, the shape of the binding site is extensively remodeled and does not present enough space for allowing an orientation of the ligand comparable to that observed in hCA VII. This is due to the presence of the bulky W36 and L43 in place of T62 and Q69 of hCA VII, respectively, which leads to the occlusion of the central portion of the pocket, and the presence of E45 in place of D71, which is oriented toward K93 and occupies the solvent-exposed region of the pocket (Appendix A). The results obtained for hCA VII, hCA XII, hCA IX, and hCA II were then analyzed in terms of ligand–protein interaction energy, attempting to correlate the selectivity of compound **4** to the binding energies associated with the predicted ligand-protein complexes. For this purpose, the linear interaction energy (LIE) approach was employed. LIE evaluations allow the calculation of the non-bonded interactions between the ligand and the surrounding protein residues from the trajectories generated through MD simulations. Electrostatic and van der Waals energetic contributions are computed for each MD snapshot, and the obtained values are then used to retrieve the average total ligand–protein interaction energy (aLIE). The MD trajectories extracted from the last 40 ns of MD simulations were used for the calculations, for a total of 400 snapshots (with a time interval of 100 ps). As shown in Appendix A, the highest binding energy (aLIE = −33.0 kcal/mol) was predicted for the hCA VII–**4** complex, consistently with the experimental results in which compound **4** showed the highest potency for hCA VII, with a submicromolar *K*_I_. Moreover, in agreement with the low micromolar activity of the ligand for hCA XII, the binding energy predicted for the hCA XII–**4** complex (aLIE = −30.1 kcal/mol) was the second highest among those estimated for the four different ligand–protein complexes. Finally, the binding energies calculated for compound **4** in complex with hCA IX and II (aLIE = −27.3 and −25.1 kcal/mol, respectively) were found to be at least 5.7 kcal/mol lower than those associated with the hCA VII–**4** complex. Notably, the lower binding energy of these two complexes appears to be primarily due to a lower electrostatic contribution, which is consistent with the binding mode predicted for the ligand into hCA IX and II, in which no H-bond is formed by the phenolic ring of the ligand, as instead observed in the hCA VII–**4** and hCA XII–**4** complexes. These results further contribute to rationalizing the SAR data obtained for compound **4** from a quantitative point of view.

In conclusion, the results of our modeling studies enabled a rationalization of the SAR data that emerged from the experimental results, suggesting a reliable binding mode of compound **4** within hCA VII, as well as within hCA II, IX, and XII, in agreement with the different potencies of the ligand against the four isoforms, and potentially justifying the low activity of the compound against hCA I, VA, and VB.

#### 2.3.3. Physicochemical and Pharmacokinetic Property Calculations

The lipophilicity of compounds **1–7** was previously evaluated in [73]. In the present work, a series of parameters affecting the drug-likeness and bioavailability of the studied *E*-stilbenol derivatives **1–9** was calculated using QikProp to complete their physicochemical profiles and to evaluate if there is an improvement in the pharmacokinetic properties of synthesized compounds compared to RSV (Table 5) [103]. Based on their calculated physicochemical and pharmacokinetic properties, all compounds showed a drug-like profile. Only compound **4** presents a slightly elevated lipophilicity value of 5.053 (logP_oct/water_ > 5). All compounds (except for **6**) possess a high value for predicted apparent MDCK cell permeability that is a good mimic for the blood–brain barrier permeability, but the values of predicted brain–blood partition coefficient are too high (−3.0 < QPlogBB < −1.2). All compounds present excellent oral absorption (human oral absorption and percent of human oral absorption) and remarkable cell permeability (QPPCaco). Overall, the pharmacokinetic properties of these synthetic derivatives are improved compared to those of RSV, particularly their abilities to reach the CNS.

## 3. Materials and Methods

### 3.1. Chemistry

The synthesis of compounds **1–9** was carried out following reported procedures [78,81]. The appropriate 4-hydroxybenzaldehyde and aryl acetic acid were mixed in the presence of piperidine at 130 °C. After aqueous work-up and purification on silica gel column chromatography, the desired phenols were obtained in high purity, as confirmed by spectroscopical experiments.

### 3.2. Biology

#### 3.2.1. In Vitro MAO Inhibition Assay

Activity measurements for MAO-A and MAO-B were carried out as reported in the literature [104,105]. Recombinant human MAO-A and MAO-B were obtained commercially (Sigma-Aldrich, Saint Louis, MO, USA) and were used as enzyme sources. Kynuramine is a non-specific substrate and served as the substrate for both MAO isoforms. The MAOs metabolize kynuramine to produce 4-hydroxyquinoline, which was measured at the endpoint of the enzyme reactions by fluorescence spectrophotometry. By evaluating the fluorescence of 4-hydroxyquinoline (~3–6 µM) in the absence and presence of the test inhibitors (1 and 10 µM), it was established that, under the experimental conditions, resveratrol and the analogs evaluated here did not fluoresce or quench the fluorescence of 4-hydroxyquinoline.

#### 3.2.2. In Vitro CA Inhibition Assay

The CA-catalyzed CO_2_ hydration activity was determined on an Applied Photophysics stopped-flow instrument (SX.18MV-R, Headquarters Applied Photophysics Limited, 21 Mole Business Park, Leatherhead, Surray, KT22 7BA, United Kingdom) [106] using phenol red at a concentration of 0.2 mM as a pH indicator with 20 mM HEPES (pH 7.5) as the buffer and 20 mM Na_2_SO_4_ (for maintaining constant the ionic strength), following the initial rates of the CA-catalyzed CO_2_ hydration reaction for a period of 10–100 s and working at the maximum absorbance of 557 nm. The CO_2_ concentrations ranged from 1.7 to 17 mM for the determination of the kinetic parameters and inhibition constants. For each inhibitor, six traces of the initial 5–10% of the reaction were used in order to determine the initial velocity. The uncatalyzed reaction rates were determined in the same manner and subtracted from the total observed rates. Stock solutions of inhibitor (0.1 mM) were prepared in distilled water, and dilutions up to 0.01 nM were prepared. Solutions containing inhibitor and enzyme were preincubated for 15 min at room temperature prior to assay in order to allow the formation of the E−I complex. The inhibition constants were obtained as nonlinear least-squares protocols using PRISM 3 and are the mean from at least three different measurements. All CAs are recombinant and were obtained in-house, following the procedure briefly described as reported earlier [107,108].

*Escherichia coli* BL21 (DE3) cells transformed with the appropriate plasmid resulted in the production of the recombinant hCAs as a fusion protein containing a His-tag tail at its N-terminus. After sonication and centrifugation, most of the CA activity was recovered in the soluble fraction of the *E. coli* cell extract. Using an affinity column (His-select HF (High Flow) nickel affinity gel), the appropriate hCA was purified to obtain the enzyme at least 95% purity.

### 3.3. Computational Studies

#### 3.3.1. In Silico Studies on Monoamine Oxidases

##### Molecular Docking

Molecular modeling experiments were performed on Schrödinger Life-Sciences Suite 2021–4. Maestro (v13) [84]. The three-dimensional X-ray structures of MAO-A and MAO-B were obtained from RCSB Protein Data Bank (PDB ID: 2Z5X and 2V5Z, respectively). Protein Preparation Wizard in Maestro was used to fix, optimize, and minimize the crystal structures [109]. Binding sites were analyzed using the SiteMap module of Maestro [110,111]. SiteScore, Dscore, size, and volume of the sites were calculated using the default parameter. All the ligands were drawn as 2D structures from Maestro and prepared using LigPrep to find all possible tautomers and protonation states at pH 7.0 ± 0.4 with Epik [112,113]. Water molecules more than 3 Å away from the FAD and forming fewer than 2 H-bonds were removed. The Glide Grids were created by using the center of mass of crystallographic ligands. Rotatable OH/SH groups were defined for Cys323, Tyr407, and Tyr444 in MAO-A and Cys172, Tyr398, and Tyr435 in MAO-B. Molecular docking analyses were performed using Glide software [114,115]. The SP docking protocol was used by setting 5000 poses per ligand for the initial phase and 400 poses per ligand for energy minimization with the OPLS4 forcefield. The reliability of the docking protocol was tested by the docking analysis of the crystallographic ligands.

##### Molecular Dynamics and MM-GBSA Calculation

MD simulation was carried out using Desmond, available in the Schrödinger Suite 2021–4 [84]. The complexes of MAO-B with the docked poses of compounds **4** and **5** and of MAO-A with the docked pose of compound **4** were embedded in an orthorhombic box of TIP4P water molecules resulting in systems of 61,986, 61,680, and 86,342 atoms, respectively. In order to balance the system charge, four Na ions were added to both MAO_B complexes, and two Na ions were added to MAO-A:**4** system. Six relaxation stages were applied to the systems as a default protocol before the simulation. The systems were treated with the OPLS4 force field, a normal pressure–temperature (NPT) ensemble with a Nose–Hoover thermostat set to 300 K and a Martyna–Tobias–Klein barostat set to 1.01325 bar pressure. The simulation’s production phase lasted 100 ns, recording frames every 100 ps. The smooth particle mesh Ewald method was used to examine the electrostatic interactions. For the MM-GBSA calculation, the trj_parch.py python tool was exploited to reduce the number of water molecules in each complex extracted from the trajectory, sample 20 representative frames along the simulation, and align the resulting structures. Sampled frames were exploited to calculate the MM-GBSA energy using the thermal_mmgbsa.py command.

#### 3.3.2. In Silico Studies on Carbonic Anhydrases

##### Molecular Docking

The crystal structures of hCA I (PDB code 1AZM) [116], hCA II (PDB code 4E3H) [100], hCA VII (PDB code 3MDZ), hCA IX (PDB code 5FL4) [117], and hCA XII (PDB code 1JCZ) [118] were taken from the Protein Data Bank [119], while for hCA VA and hCA VB, previously developed homology models were used [120,121]. Automated docking was carried out for compound 4 by means of the GOLD 5.1 program [122,123], using the PLP scoring function. The region of interest for the docking studies contained all residues within 15 Å from the largest bound ligand among all reference X-ray structures. The “allow early termination” command was deactivated, while the possibility for the ligand to flip ring corners was activated. The remaining GOLD default parameters were used, and the ligand was submitted to 100 genetic algorithm runs. The docking solutions were clustered using an RMS threshold of 2.0 Å, and the best docked conformation was taken into account.

##### Molecular Dynamics Simulations

All molecular dynamics (MD) simulations were performed with AMBER, version 20 [124]. Each complex was subjected to an MD procedure based on an already successfully applied protocol, using the ff14SB force field at 300 K [125]. Prior to MD simulations, each complex was placed in a rectangular parallelepiped water box and solvated with a 15 Å water cap using the TIP3P explicit solvent model for water. Sodium or chlorine ions were then added as counterions for the neutralization of the solvated system. Each system was subjected to two stages of energy minimization, each composed of 5000 steps of steepest descent followed by conjugate gradient until a convergence of 0.05 kcal/(mol·Å^2^) was reached. In the first stage, the whole protein was blocked with a position restraint of 500 kcal/(mol·Å^2^) to uniquely minimize the position of the water molecules, while in the second stage, the entire system was energy-minimized by applying a harmonic potential of 10 kcal/(mol·Å^2^) only to the protein α carbons. The minimized complexes were then used as the starting point for a total of 50 ns of MD simulation. A 0.5 ns constant-volume simulation, in which the temperature of the system was raised from 0 to 300 K, was initially performed. In the second step, the system was equilibrated through a 3 ns constant-pressure simulation, maintaining the temperature at the constant value of 300 K with the use of a Langevin thermostat. An additional 46.5 ns of constant-pressure MD was then performed, for a total of 50 ns of simulation. In all three MD steps, a harmonic potential of 10 kcal/(mol·Å^2^) was applied to the protein α carbons. All simulations were performed using particle mesh Ewald (PME) electrostatics with a cutoff of 10 Å for non-bonded interactions and periodic boundary conditions. A simulation step of 2.0 fs was employed, as all bonds involving hydrogen atoms were kept rigid using the SHAKE algorithm. General Amber force field (GAFF) parameters were used for the ligand, whose partial charges were calculated with the AM1-BCC method as implemented in the Antechamber suite of AMBER 20.

##### Binding Energy Evaluation

The linear interaction energy (LIE) was calculated between the ligand and the surrounding protein residues lying within a 12 Å radius from it, as previously performed [126]. The ccptraj module of AMBER 20 was used for the calculations, employing the trajectories extracted from the last 40 ns of MD simulations, for a total of 400 snapshots (with a time interval of 100 ps). The average LIE values (aLIE) were obtained as the sum of the average electrostatic (EELE) and van der Waals (EVDW) energy contributions expressed as kcal/mol.

#### 3.3.3. Physicochemical and Pharmacokinetic Property Calculations

Synthesized ligands **1**–**9** and RSV structures were submitted to QuikProp calculation applying default settings in Maestro [84].

## 4. Conclusions

The pathogenesis of NDs consists of a complex series of aspects including aging, lifestyle, and genetic factors. Multi-target designed ligands (MTDLs) are a modern approach that may provide effective pharmacological responses by acting at diverse receptors or enzymatic systems involved in the etiopathogenesis. We aimed to study dual RSV-based inhibitors of the MAOs and hCAs, with both enzymes being involved in different neurological disorders. All RSV derivatives inhibit the MAOs in the low micromolar range (0.43–21.3 µM for MAO-A and 0.011 to 14.2 µM for MAO-B), showing higher potency than RSV, while MAO-B is selectively inhibited. Docking studies suggested that the introduction of trifluoromethyl in the 4-position and a chlorine atom in the 3′-position leads to a significant increase in the inhibition activity (compound **4**). Regarding the inhibitory activity on hCAs, the preliminary structure–activity relationships revealed that only compounds with a chlorine atom in the 3′-position possess low *K*_I_ values in the low micromolar range. Moreover, among them, compound **4** has demonstrated effective inhibition abilities which is due to the presence of 4-CF_3_ in aromatic ring A. This group is the cause of the establishment of effective bonding interactions with the catalytic site in a different orientation compared to the classic sulfonamide inhibitors, opening interesting opportunities for future optimization. In fact, even if its inhibition is lower with respect to references, this compound is distinguished by its ability to bind both studied receptors through favorable structural characteristics. Lastly, the ADME prediction results indicated that all compounds showed promising drug-like properties with a view to biological action at the CNS level.

In conclusion, the exploration of new RSV-based molecules gives important insights into the design of novel dual compounds that would be retained for further research. Compound **4** emerged as a promising advanced compound that addressed multiple factors associated with NDs. After the screening, compound **4** with the highest inhibition ability was selected as a candidate for further studies.

## Figures and Tables

**Figure 1 molecules-27-07816-f001:**
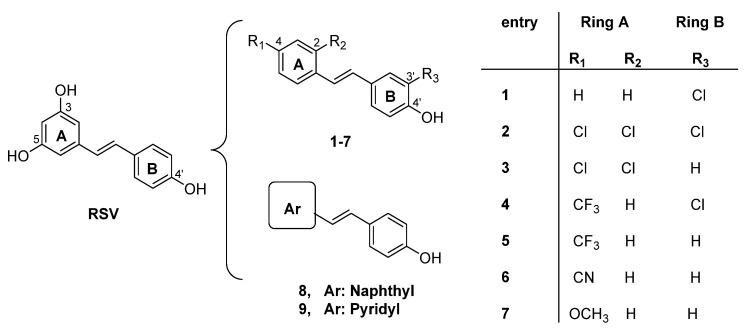
Resveratrol (RSV) and studied derivatives **1**–**9**.

**Figure 2 molecules-27-07816-f002:**
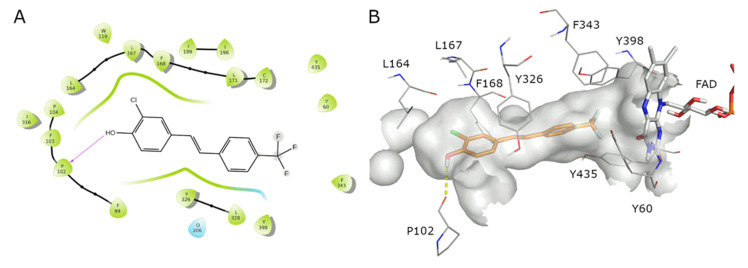
The catalytic site of MAO-B in complex with **4**. (**A**) The 2D ligand interaction diagram; (**B**) the best docked pose of ligand **4** (orange sticks) with the surrounding residues of MAO-B shown (grey lines).

**Figure 3 molecules-27-07816-f003:**
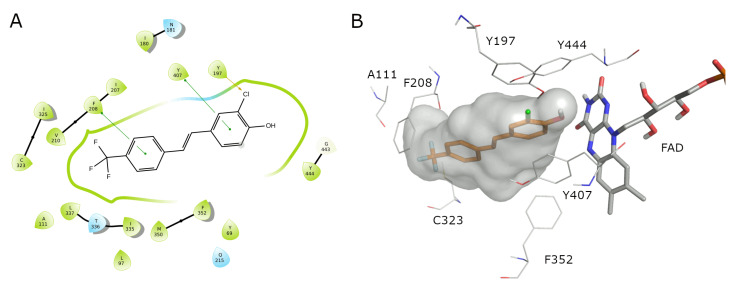
The catalytic site of MAO-A in complex with **4**. (**A**) The 2D ligand interaction diagram; (**B**) the best docked pose of ligand **4** (orange sticks) with the surrounding residues of MAO-A shown (grey lines).

**Figure 4 molecules-27-07816-f004:**
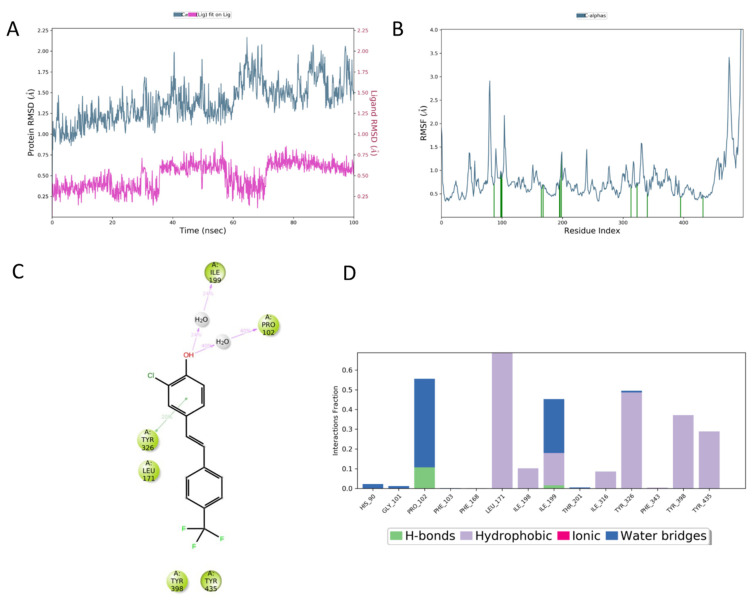
(**A**) RMSD evolution during the MD simulation of the protein (light blue, calculated on the C-alpha, left *Y*-axis) and the ligand (compound **4**) (pink, calculated on all atoms, right *Y*-axis). (**B**) Protein RMSF; green lines indicate residues interacting with ligand **4**. (**C**) The 2D representation of most conserved ligand–protein interactions with the indication of the persistence (%) along the simulation; (**D**) depiction of frequency and type of ligand–protein interaction along with the MD simulation. Ligand **4** forms mainly hydrophobic contacts with the protein.

**Figure 5 molecules-27-07816-f005:**
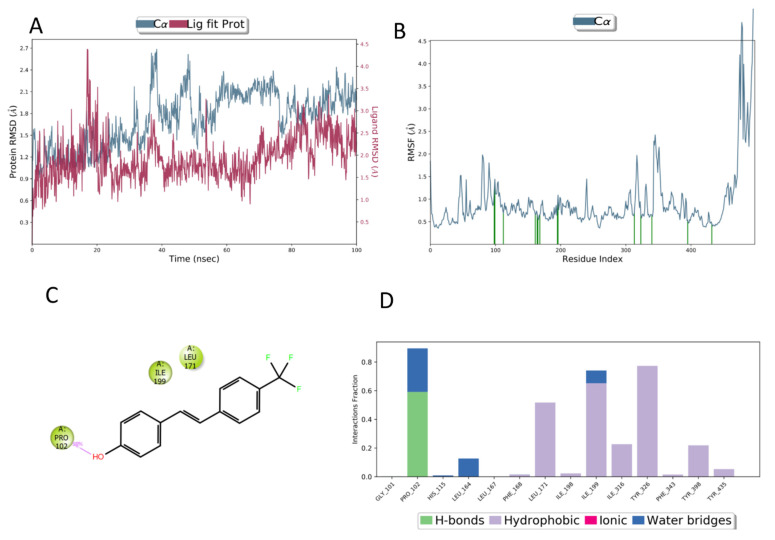
(**A**) RMSD evolution during the MD simulation of the protein (light blue, calculated on the C-alpha, left *Y*-axis) and the ligand (compound **5**) (purple, calculated on all atoms, right *Y*-axis); (**B**) protein RMSF, green lines indicate residues involved in interactions with ligand **5**; (**C**) 2D representation of most conserved ligand–protein interactions with the indication of the persistence (%) along the simulation; (**D**) depiction of the frequency and type of ligand–protein interaction along with the MD simulation. Ligand **5** forms a stable H-bond with Pro102.

**Figure 6 molecules-27-07816-f006:**
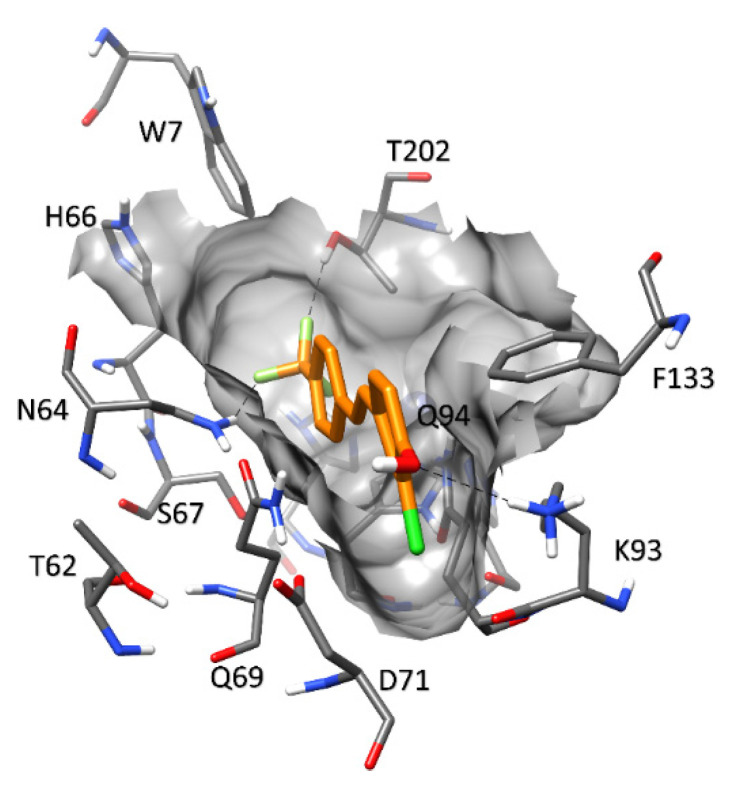
Predicted binding mode of compound **4** in hCA VII.

**Figure 7 molecules-27-07816-f007:**
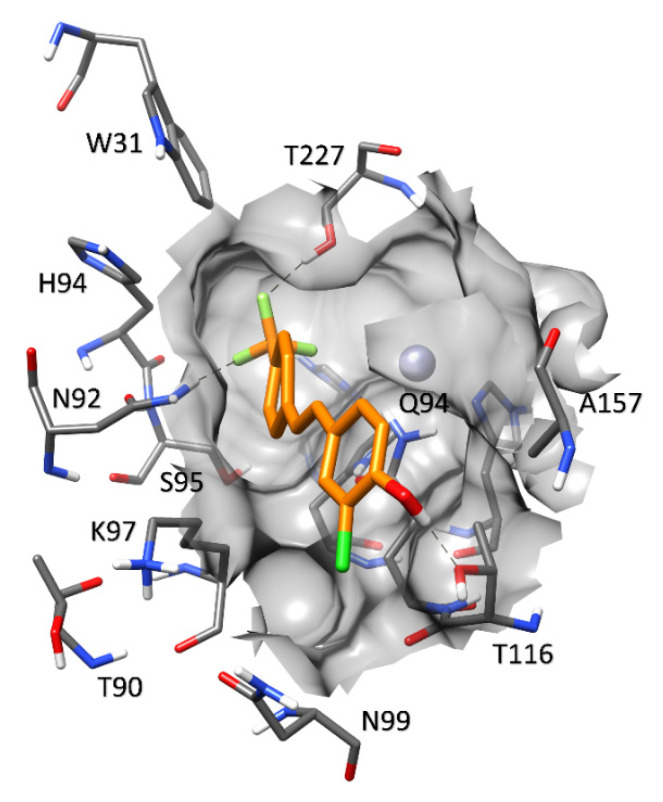
Predicted binding mode of compound **4** in hCA XII.

**Figure 8 molecules-27-07816-f008:**
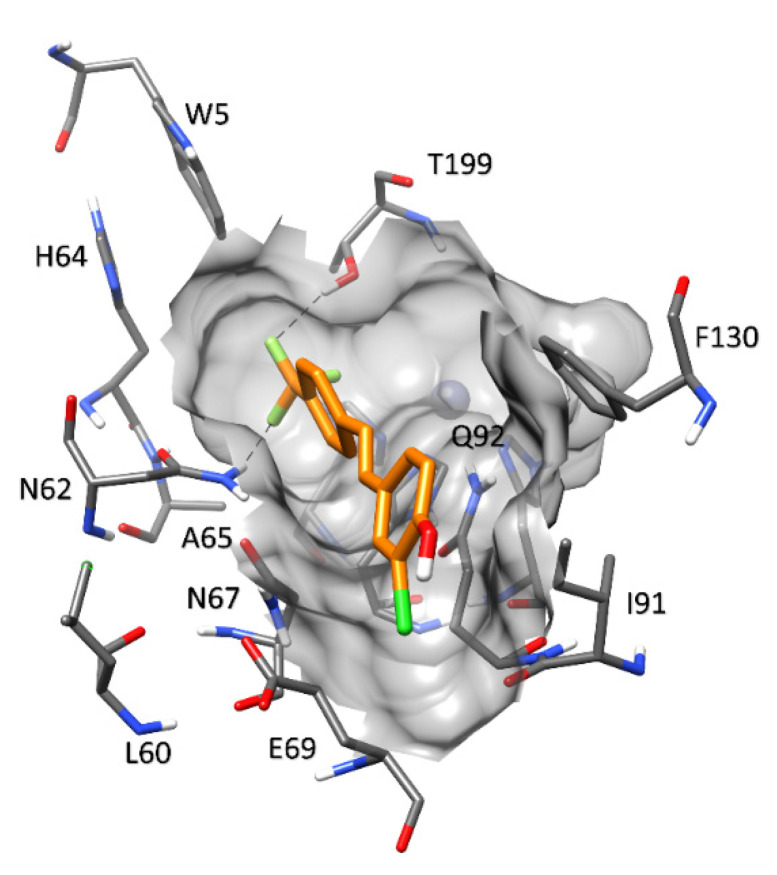
Predicted binding mode of compound **4** in hCA II.

**Figure 9 molecules-27-07816-f009:**
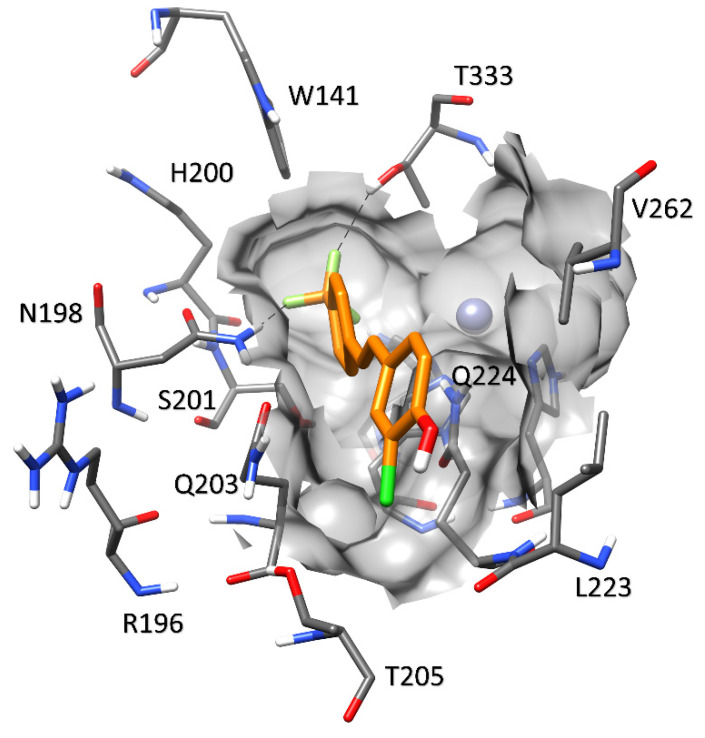
Predicted binding mode of compound **4** in hCA IX.

**Table 1 molecules-27-07816-t001:** The MAO inhibitory activities * of compounds **1–9** and RSV.

Compound	MAO-A IC_50_ ± S.D. (µM)	MAO-B IC_50_ ± S.D. (µM)
**1**	3.06 ± 0.100	0.156 ± 0.009
**2**	2.02 ± 0.175	1.44 ± 0.035
**3**	1.79 ± 0.095	1.52 ± 0.067
**4**	0.433 ± 0.127	0.011 ± 0.0065
**5**	9.77 ± 0.553	14.2 ± 1.82
**6**	2.01 ± 0.033	0.103 ± 0.003
**7**	2.71 ± 0.016	0.185 ± 0.009
**8**	2.20 ± 0.093	0.387 ± 0.022
**9**	21.3 ± 7.30	11.9 ± 1.51
**RSV**	13.5 ± 1.28	>100
**Harmine**	0.0041 ± 0.00007	-
**Isatin**	8.43 ± 0.245	3.90 ± 0.792

* Data are given as the mean ± S.D. of three independent experiments. RSV: resveratrol.

**Table 2 molecules-27-07816-t002:** The *K*_I_ values for the inhibition of hCA I, II, V, VII, IX, and XII by derivatives **1**–**9**, RSV, and AAZ as reference inhibitor.

*K*_I_ (µM) *							
Compound	hCA I	hCA II	hCA VA	hCA VB	hCA VII	hCA IX	hCA XII
**1**	89.5	35.8	9.0	84.5	>100	23.6	2.7
**2**	>100	>100	37.2	32.2	4.2	81.1	4.5
**3**	>100	>100	86.0	>100	7.5	>100	>100
**4**	86.8	15.6	70.9	36.2	0.7	17.4	6.9
**5**	>100	43.4	83.0	8.7	>100	94.6	>100
**6**	>100	75.3	>100	9.5	7.4	>100	>100
**7**	>100	>100	>100	9.7	9.4	81.8	>100
**8**	>100	77.5	98.2	21.1	>100	47.7	>100
**9**	78.4	11.8	>100	>100	>100	75.3	>100
**RSV**	2.2	2.8	4.7	4.6	4.3	0.8	0.9
**AAZ**	0.250	0.012	0.063	0.054	0.002	0.026	0.006

* Data are the mean of three independent experiments, conducted by a stopped-flow technique (errors were in the range of ±5–10% of the reported values). RSV: resveratrol; AAZ: acetazolamide.

**Table 3 molecules-27-07816-t003:** SiteMap analysis of the *E*-stilbenol ligands.

Protein	SiteScore	Dscore	Size	Volume	Phobic	Philic	Balance
**MAO-A**	1.17	1.191	130	240.958	3.474	0.882	3.939
**MAO-B**	1.209	1.231	163	270.37	3.12	0.913	3.418

SiteScore: drug-binding site >0.8; Dscore: druggability score; size: number of site points; volume: Å^3^; phobic: the hydrophobic character of the site (>1); philic: the hydrophilic character of the site (>1); balance: ratio of hydrophobic and hydrophilic (>1.6).

**Table 4 molecules-27-07816-t004:** MM-GBSA calculated binding energies for the frames extracted from the MD simulation of the three complexes.

Complex	MMGBSA dG Bind ^a^	MMGBSA dG Bind Coulomb ^b^	MMGBSA dG Bind Covalent ^c^	MMGBSA dG Bind Hbond ^d^	MMGBSA dG Bind Lipo ^e^	MMGBSA dG Bind Packing ^f^	MMGBSA dG Bind Solv GB ^g^	MMGBSA dG Bind vdW ^h^
MAO-B:**4**	−24.09	−12.15	2.01	−0.64	−28.66	−5.49	69.83	−49.00
MAO-B:**5**	−21.00	−16.04	2.01	−0.61	−25.35	−3.51	64.79	−42.29
MAO-A:**4**	−23.89	−12.55	2.00	−0.61	−24.84	−4.24	59.68	−43.33

^a^ Prime calculated binding DG; ^b^ Coulomb energy; ^c^ covalent binding energy; ^d^ hydrogen-bonding correction; ^e^ lipophilic energy; ^f^ π–π packing correction; ^g^ generalized Born electrostatic solvation energy; ^h^ van der Waals energy.

**Table 5 molecules-27-07816-t005:** Physicochemical and pharmacokinetic properties of the *E*-stilbenol ligands.

ID	mol MW	accptHB	donorHB	CIQPlogS	Human Oral Absorption	Percent Human Oral Absorption	QPPCaco	QPlogHERG	QPPMDCK	QPlogBB	CNS	QPlogKhsa	# metab
**1**	230.693	0.75	1	−4.217	3	100	3434.609	−5.475	4398.946	0.049	1	0.395	1
**2**	299.583	0.75	1	−5.662	3	100	3434.792	−5.324	10,000	0.357	1	0.634	1
**3**	265.138	0.75	1	−4.938	3	100	3014.508	−5.372	8693.74	0.146	1	0.513	1
**4**	298.692	0.75	1	−5.643	3	100	3434.791	−5.506	10,000	0.314	1	0.673	1
**5**	264.247	0.75	1	−4.920	3	100	3014.399	−5.557	7150.372	0.102	1	0.552	1
**6**	221.258	2.25	1	−4.491	3	94.309	625.083	−5.691	297.703	−0.916	−1	0.149	1
**7**	226.274	1.5	1	−3.858	3	100	3014.378	−5.471	1630.445	−0.236	0	0.314	2
**8**	246.308	0.75	1	−4.783	3	100	3014.399	−6.226	1630.457	−0.187	0	0.667	1
**9**	197.236	2.25	1	−3.011	3	100	1629.035	−5.308	838.343	−0.407	0	−0.015	3
**RSV**	228.247	2.25	3	−3.396	3	82.354	280.757	−5.277	125.332	−1.28	−2	−0.172	3

CIQPlogS: conformation-independent predicted aqueous solubility, log S; S in mol dm^−3^ is the concentration of the solute in a saturated solution (−6.5/+0.5). HumanOralAbsorption: predicted qualitative human oral absorption, 1 for low, 2 for medium, 3 for high. PercentHumanOralAbsorption: predicted human oral absorption on a 0 to 100% scale. QPPCaco: predicted apparent Caco-2 cell permeability in nm/s; Caco-2 cells are a model for the gut–blood barrier (<25 poor, >500 great). QPlog-HERG: predicted IC_50_ value for the blockage of HERG K^+^ channels (concern below −5). QPPMDCK: predicted apparent MDCK cell permeability in nm/s; MDCK cells are considered to be a good mimic for the blood–brain barrier (<25 poor, >500 excellent). QPlogBB: predicted brain–blood partition coefficient (−3.0/+1.2). CNS: predicted central nervous system activity on a −2 (inactive) to +2 (active) scale. QPlogKhsa: prediction of binding to human serum albumin (−1.5/+1.5). #metab: number of likely metabolic reactions.

## Data Availability

Data are available within the manuscript.

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
