# Peer review of "Resveratrol Analogues as Dual Inhibitors of Monoamine Oxidase B and Carbonic Anhydrase VII: A New Multi-Target Combination for Neurodegenerative Diseases?"

_molecules, 2022, doi:10.3390/molecules27227816_

Round 1
Reviewer 1 Report
This manuscript reports experimental studies of synthetized resveratrol analogues on the inhibition of MAO and carbonic anhydrases to be used as multi-targeted directed ligand strategy. It uses molecular docking to explain the experimental results. These results can contribute to increase the knowledge in this field. A number of questions should be addressed and revised accordingly:
1. Data of IC50 of MAO are given. If possible, it would be better to give ki values and the concentration of substrate should be added. The IC50 of harmine is lower than that described in the literature.
2. No experimental results are given on the type of binding and type of nhibition of MAO. The authors analyze by docking the compounds assuming that they inhibit the enzyme by a competitive mode. However, to assume that they should accomplish experimental studies on kinetics (competitive inhibition) and type of binding (reversibility). The same could be considered for CA enzymes.
3. In L168: Previous research on the inhibition of MAO by resveratrol and analogues could be added such as BBRC (2006) 344, 688-695; J. Chromatogr. B (2018), 1073, 136-144; Biorganic Chem. (2020), 103707.
4. Table 3 (sitemap analysis of E-stilbene ligands): please, explain how these results were obtained. The computational studies on MAO and CA (page 6-15) are descriptive explaining and discussing the experimental results of Tables 1 and 2. Perhaps, it could be good to include some selected numerical results of docking that led to those explanations. The computational analysis of MAO and CA might be better summarized and focused on the selected compounds and enzymes with more activity.
5. The assay of MAO to obtain results in Table 1 should be described in more detail. Natural compounds including resveratrol, that is an inhibitor of MAO-A and a poor inhibitor of MAO-B (J. Chromatogr. B (2018), 1073, 136-144; Food Chem. Toxicol. (2011), 49, 1773-1781) could interfere with the analysis of MAO activity. A chromatographic method has been proposed to solve this. Direct fluorescence is used here for detection; then, could the synthesized fluorescent derivatives of resveratrol somehow interfere with 4-hydroxyquinoline detected by fluorescence? This topic should be discussed.
6. The assay of carbonic anhydrases should be described in more detail to be reproduced by others. In addition, a reference on the reliability of this assay should be included.
7. A brief description on the synthesis and purity of the compounds used, including isomer configuration with an appropriate reference should be included. Also, on how the recombinant enzymes of carbonic anhydrates were obtained and with appropriate references.
8. References on the x-ray structures of MAO used here structural analysis of the active site are missing (e.g for MAO, J. Med. Chem. (2004) 47, 1767-1774; Nat Struct Biol. (2002), 9, 22-26; Febss Lett. (2004), 564, 225-228, PNAS (2003), 100, 9750-9755; PNAS (2008), 105, 5739–5744; J Mol Biol. (2004), 338, 103–114; PNAS (2005), 102, 12684–12689. The same for CA enzymes.
9. In results (L296-311) there appears to be some wrong assignation of Figures 5 (4B?), Figure 6 and Figure 7 (Figure 5?). The description of the captions are not including the isozyme.
10. The English should be revised
Author Response
Thanks to the reviewers for their suggestions.
In agreement with the reviewer's comment, we revised the manuscript and corrected the mistakes.
All the modifications have been evidenced in red in the manuscript and indicated below.
References have been improved and the new references have been written in red.
Referee 1
This manuscript reports experimental studies of synthetized resveratrol analogues on the inhibition of MAO and carbonic anhydrases to be used as multi-targeted directed ligand strategy. It uses molecular docking to explain the experimental results. These results can contribute to increase the knowledge in this field. A number of questions should be addressed and revised accordingly:
- Data of IC50 of MAO are given. If possible, it would be better to give ki values and the concentration of substrate should be added. The IC50 of harmine is lower than that described in the literature.
In this study, the MAO inhibition potencies are reported as IC50 values. IC50 values are more practical to measure and are widely accepted as measures of potency in medicinal chemistry projects. The measurement of Ki values requires more extensive experimentation and are usually reserved for studies with only a few compounds. However, Ki may be calculated from IC50 according to the following equation: Ki = IC50/(1 + [S]/Km) with [S] = 50 µM (Cheng & Prusoff, 1973). Km (kynuramine) equals 16.1 µM for human MAO-A and 22.7 µM for human MAO-B (Strydom et al., 2010). In our opinion, no additional value would be added to the manuscript by calculating and reporting Ki values.
- Cheng, Y. C.; Prusoff, W. H. Biochem. 1973, 22, 3099.
- Strydom, B.; Malan, S. F.; Castagnoli, N.; Bergh, J. J.; Petzer, J. P. Bioorg. Med. Chem. 2010, 18, 1018e1028.
Literature reports a Ki value for harmine of 0.016 µM for human MAO-A while our study finds an IC50 of 0.0041 µM (Reniers et al., 2011). Although not equal, the potencies are in a similar range, in the lower nanomolar concentrations.
- Reniers, J.; Robert, S.; Frederick, R.; Masereel, B.; Vincent, S.; Wouters, J. Bioorg Med Chem. 2011, 19.
- No experimental results are given on the type of binding and type of inhibition of MAO. The authors analyze by docking the compounds assuming that they inhibit the enzyme by a competitive mode. However, to assume that they should accomplish experimental studies on kinetics (competitive inhibition) and type of binding (reversibility).
The same could be considered for CA enzymes.
The study compounds do not have any functional groups that could result in irreversible inactivation of the MAO enzymes. It is thus reasonable to suggest that the study compounds are reversible MAO inhibitors. Furthermore, resveratrol has been reported to inhibit MAO “with a competitive time-independent mechanism” (Zhang et al., 2019).
To address this, the following was added to the manuscript: “The resveratrol analogues of this study do not contain functional groups that could result in irreversible inactivation of the MAO enzymes. It is thus reasonable to suggest that the study compounds are reversible MAO inhibitors. Furthermore, resveratrol has been reported to inhibit MAO with a competitive mechanism (Zhang et al., 2019).”
- Zhang, Z.; Hamada H.; Gerk PM. Biomed Res Int. 2019, 8361858.
Regarding the mechanism of CAs inhibition, nowhere in the text is it mentioned that compound 4 acts as a competitive inhibitor on the enzyme. In addition, from the docking results the compound binds to a site close to the active site acting as a non-competitive inhibitor.
- In L168: Previous research on the inhibition of MAO by resveratrol and analogues could be added such as BBRC (2006) 344, 688-695; J. Chromatogr. B (2018), 1073, 136-144; Biorganic Chem. (2020), 103707.
We agree with this suggestion and we have added these references to the text.
- Table 3 (sitemap analysis of E-stilbene ligands): please, explain how these results were obtained. The computational studies on MAO and CA (page 6-15) are descriptive explaining and discussing the experimental results of Tables 1 and 2. Perhaps, it could be good to include some selected numerical results of docking that led to those explanations. The computational analysis of MAO and CA might be better summarized and focused on the selected compounds and enzymes with more activity.
We thank the reviewer for this comment. The SiteMap analysis was performed on the MAO-A and MAO-B binding site to characterize their binding pockets. The program, in fact, maps the binding site evaluating its dimension and properties (hydrophobic and hydrophilic regions) and computes a series of parameters such as those we reported in Table 3. This analysis was aimed to get numbers that help eliciting the differences between the two enzymes’ pockets.
Additional explanation comments have been added in the manuscript to address the reviewer’s comment.
In order to obtain reliable quantitative data related to the predicted binding modes, we performed ligand-protein binding energy evaluations based on the molecular dynamics results of the predicted complexes. To this aim, linear interaction energy (LIE) evaluations calculated between the ligand and the surrounding protein residues were performed. In compliance with the Reviewer’s comment, the results of this analysis, which were consistent with the qualitative results obtained, as well as with the SAR data, were described within the manuscript. In particular, at line 514-537 of the revised manuscript, the following text has been added: “The results obtained for hCA VII, hCA XII, hCA IX and hCA II were then analyzed in terms of ligand-protein interaction energy, attempting to correlate the selectivity of compound 4 to the binding energies associated with the predicted ligand-protein complexes. For this purpose, the linear interaction energy (LIE) approach was employed. LIE evaluations allow the calculation of the non-bonded interactions between the ligand and the surrounding protein residues from the trajectories generated through MD simulations. Electrostatic and van der Waals energetic contributions are computed for each MD snapshot and the obtained values are then used to retrieve the average total ligand-protein interaction energy (aLIE). The MD trajectories extracted from the last 40 ns of MD simulations were used for the calculations, for a total of 400 snapshots (with a time interval of 100 ps). As shown in Table S1, the highest binding energy (aLIE = -33.0 kcal/mol) was predicted for hCAVII-4 complex, consistently with the experimental results in which compound 4 showed the highest potency for hCAVII, with a submicromolar KI. Moreover, in agreement with the low micromolar activity of the ligand for hCA XII, the binding energy predicted for hCA XII-4 complex (aLIE = -30.1 kcal/mol) was the second highest among those estimated for the four different ligand-protein complexes. Finally, the binding energies calculated for compound 4 in complex with hCA IX and II (aLIE = -27.3 and -25.1 kcal/mol, respectively) were found to be at least 5.7 kcal/mol lower than those associated with hCAVII-4 complex. Notably, the lower binding energy of these two complexes appears to be primarily due to a lower electrostatic contribute, which is consistent with the binding mode predicted for the ligand into hCA IX and II, in which no H-bond is formed by the phenolic ring of the ligand, as instead observed in hCAVII-4 and hCAXII-4 complexes. These results further contribute to rationalize the SAR data obtained for compound 4 from a quantitative point of view.” Moreover, the new paragraph “3.3.2.3 Binding Energy Evaluation”, describing the methods related to the ligand-protein binding energy evaluations, has been added at lines 689-696 of the revised manuscript.
- The assay of MAO to obtain results in Table 1 should be described in more detail. Natural compounds including resveratrol, that is an inhibitor of MAO-A and a poor inhibitor of MAO-B (J. Chromatogr. B (2018), 1073, 136-144; Food Chem. Toxicol. (2011), 49, 1773-1781) could interfere with the analysis of MAO activity. A chromatographic method has been proposed to solve this. Direct fluorescence is used here for detection; then, could the synthesized fluorescent derivatives of resveratrol somehow interfere with 4-hydroxyquinoline detected by fluorescence? This topic should be discussed.
While the MAO assay has been described in detail in the literature references provided in the experimental, the following was added to the experimental:
“By evaluating the fluorescence of 4-hydroxyquinoline (~3-6 µM) in the absence and presence of the test inhibitors (1 and 10 µM) it was established that, under the experimental conditions, resveratrol and the analogues evaluated here did not fluoresce or quench the fluorescence of 4-hydroxyquinoline.”
- The assay of carbonic anhydrases should be described in more detail to be reproduced by others. In addition, a reference on the reliability of this assay should be included.
The assay was described more accurately, and the reliability reference was added.
- A brief description on the synthesis and purity of the compounds used, including isomer configuration with an appropriate reference should be included.
We have introduced paragraph “Chemistry” in the “Results and Discussion” and in “Materials and Methods” sections. Here we briefly described the method of synthesis and inserted bibliographical references. We have also included a note about the 1H-NMR study of pure final products and the geometry of the double bond with appropriate reference.
Also, on how the recombinant enzymes of carbonic anhydrates were obtained and with appropriate references.
The method for obtaining recombinant CA enzymes was added.
- References on the x-ray structures of MAO used here structural analysis of the active site are missing (e.g for MAO, J. Med. Chem. (2004) 47, 1767-1774; Nat Struct Biol. (2002), 9, 22-26; Febss Lett. (2004), 564, 225-228, PNAS (2003), 100, 9750-9755; PNAS (2008), 105, 5739–5744; J Mol Biol. (2004), 338, 103–114; PNAS (2005), 102, 12684–12689. The same for CA enzymes.
We are grateful to the referee for the suggestion. We inserted the references for the MAO X-ray complexes we used in our study. Moreover, all available references related to the X-ray structures of CA isoforms have been added to the text.
- In results (L296-311) there appears to be some wrong assignation of Figures 5 (4B?), Figure 6 and Figure 7 (Figure 5?). The description of the captions are not including the isozyme.
Corrected accordingly.
- The English should be revised
Corrected accordingly.
Reviewer 2 Report
The paper of Carradori et al. investigates the inhibition potency of a series of resveratrol analogues towards monoamine oxidase (MAO) and carbonic anhydrase (hCA) enzymes. At least one of these analogues exhibits promising inhibitory activity towards both MAO-B and hCA VII enzyme; in turn, it features substantial selectivity among MAO and hCA variants. It has been suggested that the selected compound may be used as a dual-target inhibitor, which appears to be appealing in the context of control and prevention of neurodegenerative diseases.
The present experimental research has been extensively supported by computational treatments (docking and MD simulation), which greatly enhance the interpretation of kinetic studies and provide reasonable explanation of the observed inhibition activity and selectivity. I think the present research is of very sound quality and I recommend publication in Molecules. Nevertheless, the authors are welcome to consider the following remarks and suggestions:
(1) A general remark on the possible dual/multi target inhibition: while compound 4 exhibits highly potent (nanomolar) inhibition of MAO-B (which is in the range of established MAO inhibitors such as selegiline), inhibition potency of that same compound appears to be much poorer towards hCA VII (at least in comparison with the reference inhibitor). This caveat deserves to be stressed in the discussion.
(2) Simulation of docking of compound 4 into the active site predicts an entirely different (180 deg. rotated) orientation between MAO-A and MAO-B. How is this reflected in the estimated binding free energies? Binding energies are reported for MAO-B, but not for the A-isoform.
(3) Analysis of the docked compounds into the active sites of hCA are illustrative and provide a reach source of interpretation, but again, it would be desirable to estimate binding free energies as well – do binding free energies reasonably explain the observed inhibition activity and selectivity?
(4) Apart from numerous kinetic studies, inhibition of MAO enzymes and their mechanism of action/inhibition have been quite recently investigated by simulation, see for example: T. Tandaric, R. Vianello, ACS Chem. Neurosci. 2019, 10, 8, 3532; T. Tandaric et al., Int. J. Mol. Sci. 2020, 21, 6151; A. Prah et al., Molecules 2022, 27, 6711. Eventually, the presently reported binding free energies are in quite a good match with those reported for rasagiline and selegiline computed in one of these studies by MM-GBSA. The suggested examples deserve to be briefly discussed to further support the present results.
(5) Pg. 3, line 138, repetition “substituted substituted”.
Author Response
Thanks to the reviewers for their suggestions.
In agreement with the reviewer's comment, we revised the manuscript and corrected the mistakes.
All the modifications have been evidenced in red in the manuscript and indicated below.
References have been improved and the new references have been written in red.
Reviewer 2
The paper of Carradori et al. investigates the inhibition potency of a series of resveratrol analogues towards monoamine oxidase (MAO) and carbonic anhydrase (hCA) enzymes. At least one of these analogues exhibits promising inhibitory activity towards both MAO-B and hCA VII enzyme; in turn, it features substantial selectivity among MAO and hCA variants. It has been suggested that the selected compound may be used as a dual-target inhibitor, which appears to be appealing in the context of control and prevention of neurodegenerative diseases. The present experimental research has been extensively supported by computational treatments (docking and MD simulation), which greatly enhance the interpretation of kinetic studies and provide reasonable explanation of the observed inhibition activity and selectivity. I think the present research is of very sound quality and I recommend publication in Molecules. Nevertheless, the authors are welcome to consider the following remarks and suggestions:
- A general remark on the possible dual/multi target inhibition: while compound 4 exhibits highly potent (nanomolar) inhibition of MAO-B (which is in the range of established MAO inhibitors such as selegiline), inhibition potency of that same compound appears to be much poorer towards hCA VII (at least in comparison with the reference inhibitor). This caveat deserves to be stressed in the discussion.
The discussion and the conclusions have been improved.
(2) Simulation of docking of compound 4 into the active site predicts an entirely different (180 deg. rotated) orientation between MAO-A and MAO-B. How is this reflected in the estimated binding free energies? Binding energies are reported for MAO-B, but not for the A-isoform.
MD and following MM-GBSA calculations were carried out also on the MAO-A:4 complex. The resulting values have been inserted and discussed in the manuscript.
(3) Analysis of the docked compounds into the active sites of hCA are illustrative and provide a reach source of interpretation, but again, it would be desirable to estimate binding free energies as well – do binding free energies reasonably explain the observed inhibition activity and selectivity?
We thank the Reviewer for this suggestion. Due to the presence of the prosthetic zinc ion in the catalytic site of hCAs, we performed ligand-protein binding energy evaluations using a linear interaction energy (LIE) approach. In fact, although no direct ligand-ion interaction is predicted in the different ligand-hCA complexes, the presence of an ion in close proximity of the ligand may not allow to determine reliable binding free energies calculated with the use of implicit solvent approximations, such as GBSA and/or PBSA approaches, in which ions and solvent molecules are not explicitly taken into accounts. For this reason, the LIE approach, in which all residues (included solvent and ions) within a defined radius from the ligand are explicitly considered, was used for these evaluations. In compliance with the Reviewer’s comment, the results of this analysis, which were consistent with the qualitative results obtained, as well as with the SAR data, were described within the manuscript. at line 514-537 of the revised manuscript, the following text has been added: “The results obtained for hCA VII, hCA XII, hCA IX and hCA II were then analyzed in terms of ligand-protein interaction energy, attempting to correlate the selectivity of compound 4 to the binding energies associated with the predicted ligand-protein complexes. For this purpose, the linear interaction energy (LIE) approach was employed. LIE evaluations allow the calculation of the non-bonded interactions between the ligand and the surrounding protein residues from the trajectories generated through MD simulations. Electrostatic and van der Waals energetic contributions are computed for each MD snapshot and the obtained values are then used to retrieve the average total ligand-protein interaction energy (aLIE). The MD trajectories extracted from the last 40 ns of MD simulations were used for the calculations, for a total of 400 snapshots (with a time interval of 100 ps). As shown in Table S1, the highest binding energy (aLIE = -33.0 kcal/mol) was predicted for hCAVII-4 complex, consistently with the experimental results in which compound 4 showed the highest potency for hCAVII, with a submicromolar KI. Moreover, in agreement with the low micromolar activity of the ligand for hCA XII, the binding energy predicted for hCA XII-4 complex (aLIE = -30.1 kcal/mol) was the second highest among those estimated for the four different ligand-protein complexes. Finally, the binding energies calculated for compound 4 in complex with hCA IX and II (aLIE = -27.3 and -25.1 kcal/mol, respectively) were found to be at least 5.7 kcal/mol lower than those associated with hCAVII-4 complex. Notably, the lower binding energy of these two complexes appears to be primarily due to a lower electrostatic contribute, which is consistent with the binding mode predicted for the ligand into hCA IX and II, in which no H-bond is formed by the phenolic ring of the ligand, as instead observed in hCAVII-4 and hCAXII-4 complexes. These results further contribute to rationalize the SAR data obtained for compound 4 from a quantitative point of view. ”Moreover, the new paragraph “3.3.2.3 Binding Energy Evaluation”, describing the methods related to the ligand-protein binding energy evaluations, has been added at lines 689-696 of the revised manuscript.
(4) Apart from numerous kinetic studies, inhibition of MAO enzymes and their mechanism of action/inhibition have been quite recently investigated by simulation, see for example: T. Tandaric, R. Vianello, ACS Chem. Neurosci. 2019, 10, 8, 3532; T. Tandaric et al., Int. J. Mol. Sci. 2020, 21, 6151; A. Prah et al., Molecules 2022, 27, 6711. Eventually, the presently reported binding free energies are in quite a good match with those reported for rasagiline and selegiline computed in one of these studies by MM-GBSA. The suggested examples deserve to be briefly discussed to further support the present results.
The suggested studies have been considered in the discussion of obtained results from MM-GBSA binding DG calculations.
(5) Pg. 3, line 138, repetition “substituted substituted”.
Corrected accordingly.
Round 2
Reviewer 1 Report
The authors have addressed all the points raised before and have introduced appropriate changes in the new version. As a result the manuscript has now improved for publication.